# Melatonin Modulates Tomato Root Morphology by Regulating Key Genes and Endogenous Hormones

**DOI:** 10.3390/plants13030383

**Published:** 2024-01-27

**Authors:** Qiang Tian, Guangzheng Wang, Jianhua Dou, Yu Niu, Ruirui Li, Wangwang An, Zhongqi Tang, Jihua Yu

**Affiliations:** 1College of Horticulture, Gansu Agricultural University, Lanzhou 730070, China; tqiang12345@163.com (Q.T.); wanggz@st.gsau.edu.cn (G.W.); doujianhua1228@163.com (J.D.); 15693445082@163.com (Y.N.); 18409318527@139.com (R.L.); 173399771601@163.com (W.A.); 2State Key Laboratory of Aridland Crop Science, Gansu Agricultural University, Lanzhou 730070, China

**Keywords:** tomato roots, melatonin, hormone content, root-related gene expression

## Abstract

Melatonin plays a vital role in plant growth and development. In this study, we treated hydroponically grown tomato roots with various concentrations of exogenous melatonin (0, 10, 30, and 50 μmol·L^−1^). We utilized root scanning and microscopy to examine alterations in root morphology and cell differentiation and elucidated the mechanism by which melatonin regulates these changes through the interplay with endogenous hormones and relevant genes. The results showed that for melatonin at concentrations ranging between 10 and 30 μmol·L^−1^, the development of lateral roots were significantly stimulated, the root hair growth was enhanced, and biomass accumulation and root activity were increased. Furthermore, we elucidated that melatonin acts as a mediator for the expression of genes, such as *SlCDKA1*, *SlCYCA3;1*, *SlARF2*, *SlF3H*, and *SlKT1*, which are involved in the regulation of root morphology changes. Additionally, we observed that melatonin influences the levels of endogenous hormones, including ZT, GA3, IAA, ABA, and BR, which subsequently impact the root morphology development of tomato roots. In summary, this study shows that tomato root morphology can be promoted by the optimal concentration of exogenous melatonin (10–30 μmol·L^−1^).

## 1. Introduction

The roots vitality is apparent throughout the entire plant life cycle. The location and function of roots are pivotal for plant development and are often the primary limiting factor in nearly all natural ecosystems. Root organ evolution originates from the stem, which appears to be better suited for terrestrial life [1]. This adaptation allows plants to acquire nutrients and water from their surroundings while firmly anchoring themselves in the soil matrix [2]. Additionally, plant roots house a vascular system that supplies essential nutrients, water, and hormones to distant plant organs [3,4]. Broadly, plant roots consist of primary roots (PRs) and secondary roots. Primary roots develop during the embryonic stage, while secondary roots form after the embryo stage [5]. Moreover, secondary roots encompass lateral roots (LR) and adventitious roots (AR). Lateral roots are offshoots from primary roots, whereas adventitious roots develop on non-root tissues like hypocotyls, stems, and leaves [6,7]. Interestingly, the root structures of dicotyledonous and monocotyledonous plants diverge. Dicotyledonous plants have a vertical root system comprising central primary roots and lateral branch roots, whereas monocotyledonous plants feature a fibrous root system consisting of crown roots or adventitious roots [8]. Within the root system, there is another essential component that plays a crucial role—the root hair. Root hairs substantially increase the root surface area and boost the absorption of nutrients and water, making them of paramount importance for plant growth and development [9,10,11,12]. Prior research has demonstrated the critical role of the ion transport system on the plasma membrane of root hairs in the absorption of K^+^ and NO_3_^−^ by plants [13,14]. Simultaneously, certain genes from the *PHT1*, *AMT*, and *SULTR* families expressed in Arabidopsis root hairs participate in the absorption of phosphate [15,16], ammonium [17], and sulfate [18]. Root hair growth is influenced by numerous factors, including soil conditions, growth medium, and mineral nutrients, and it is subject to regulation by hormones such as auxin, ethylene, brassinolide, and jasmonic acid, among others [19].

Melatonin (N-acetyl-5-methoxytryptamine) was initially identified in the bovine pineal gland by Lerner in 1958. Early investigations primarily focused on melatonin as one of the foremost biomolecules in animals, notably as a neurohormone [20,21]. Researchers have established that melatonin plays a crucial role in physiological processes, including body temperature regulation, sleep, circadian rhythms, and immune system function. Additionally, it serves as a significant antioxidant in various cellular activities [22,23]. In 1995, researchers made the inaugural discovery of melatonin in plants [24,25]. Subsequently, researchers have observed the extensive presence of melatonin in the plant kingdom, spanning a minimum of 20 plant families [26]. In plants, melatonin is biosynthesized from tryptophan through a series of six enzyme-catalyzed steps [27]. Key enzymes involved in this process include tryptophan hydroxylase (TPH), tryptophan decarboxylase (TDC), tryptophan 5-hydroxylase (T5H), N-acetylhydroxyl serotonin methyltransferase (ASMT), serotonin N-acetyltransferase (SNAT), and caffeic acid O-methyltransferase (COMT). Given the resemblance in synthetic precursors and structures between melatonin and indole-3-acetic acid (IAA, auxin), numerous botanists have explored their potential roles as regulators of plant growth and development [28,29]. Hernandez-Ruiz [30] observed that melatonin treatment resulted in elongated coleoptiles in canary grass (*Phalaris canariensis*), wheat (*Triticum aestivum*), barley (*Hordeum vulgare*), and oat (*Avena sativa*). Arnao and Hernández-Ruiz [31], Posmyk et al. [32], and Wei et al. [33] reported that melatonin treatment notably enhanced the vegetative growth of maize (*Zea mays*), cucumber (*Cucumis sativus*), and *Arabidopsis*. Concurrently, prior research has demonstrated that the external application of melatonin can enhance the development of root structures in plants, including rice [34], *Arabidopsis* [35], and cucumber [36]. A number of preliminary investigations have indicated that melatonin and auxin jointly regulate root morphology [29,37]. Melatonin shares characteristics with auxin: low melatonin concentrations promote root growth, while high melatonin concentrations inhibit this process [38]. 

Tomato (*Solanum lycopersicum* L.), belonging to the Solanaceae family [39], stands as one of the world’s most crucial and extensively cultivated economic vegetable crops [40]. Previous studies have extensively investigated the impact of melatonin on the growth and development of tomato crops. Primarily, melatonin has been extensively researched for its role in enhancing tomato resistance to various abiotic stresses, encompassing drought [41,42], heavy metals [43,44], heat damage [45], cold damage [46], salt damage [47], and more. Furthermore, the melatonin treatment of tomatoes has been found to delay leaf senescence [48], boost tomato disease resistance [49,50], and improve tomato fruit quality [51,52]. Concurrently, research has revealed that melatonin treatment can induce the accumulation of nitric oxide (NO) and influence the development of adventitious roots in tomato seedlings [37]. Melatonin can stimulate hydrogen peroxide (H_2_O_2_) production and modulate the expression of cell cycle genes to facilitate lateral root development [35]. While the impact of melatonin on root morphology was first observed in tomato crops, the precise mechanism underlying how melatonin influences root morphological alterations remains elusive.

Melatonin, a relatively recent discovery, has not been thoroughly investigated for its role in elucidating root morphological alterations and the impact on growth and development through cell micromorphology, hormone content, and associated gene expression levels. In this experiment, tomato seedling roots were subjected to various melatonin concentrations via hydroponic treatment, followed by the observation of ensuing alterations in root morphology. Simultaneously, high-performance liquid chromatography (HPLC) was employed to measure the levels of other hormones in the roots of tomato seedlings, while real-time fluorescence quantitative PCR was utilized to ascertain the expression levels of genes associated with tomato roots. This approach facilitates a more in-depth understanding of the precise mechanisms by which melatonin influences the morphological changes in tomato seedling roots. The purpose of this study is to lay a foundation for the research, application, and product development of melatonin in plants.

## 2. Results

### 2.1. Effects of Exogenous Melatonin Treatment on the Fresh Weight and Root Activity of Tomato Seedlings

Following melatonin treatment at varying concentrations, the fresh weight and root activity of tomato seedlings exhibited an initial increase followed by a subsequent decrease. The fresh weight of the 50 μmol·L^−1^ treatment was notably lower than that of the other treatments, while the fresh weight of the 10 μmol·L^−1^ and 30 μmol·L^−1^ treatments showed a slight increase compared with the control (CK). However, there were no significant differences among CK, 10 μmol·L^−1^, and 30 μmol·L^−1^. Regarding root activity, the 10 μmol·L^−1^ and 30 μmol·L^−1^ treatments exhibited significantly higher levels compared with the CK and 50 μmol·L^−1^ treatments, with the 30 μmol·L^−1^ treatment displaying the highest root activity. However, there was no significant difference between the 10 μmol·L^−1^ and 30 μmol·L^−1^ treatments (Figure 1).

### 2.2. Effects of Exogenous Melatonin Treatment on Root Architecture Parameters of Tomato Seedlings

Following exogenous melatonin treatment, the root morphology of tomato seedlings exhibited consistent alterations. As illustrated in Figure 2, the length of the primary root decreased sequentially with the rising melatonin concentration. Notably, the CK, 10 μmol·L^−1^, and 30 μmol·L^−1^ treatments did not exhibit substantial differences, while the 50 μmol·L^−1^ treatment was markedly lower than the other treatments. Particularly, the number of lateral roots increased with the ascending melatonin concentration, while the length of lateral roots exhibited a gradual decrease, particularly in the 10 μmol·L^−1^ and 30 μmol·L^−1^ treatments.

Table 1 illustrates a decline in the total root length of tomato seedlings subjected to varying melatonin concentrations. The 10 μmol·L^−1^, 30 μmol·L^−1^, and 50 μmol·L^−1^ treatments exhibited lower values compared with the CK, with the 50 μmol·L^−1^ treatment showing a marginal increase compared with the 30 μmol·L^−1^ treatment. The 10 μmol·L^−1^, 30 μmol·L^−1^, and 50 μmol·L^−1^ treatments exhibited lower values compared with the CK, with the 50 μmol·L^−1^ treatment showing a marginal increase compared with the 30 μmol·L^−1^ treatment. Regarding root surface area, the only notable difference was that the 50 μmol·L^−1^ treatment exhibited a significant reduction compared with the other treatments, while there were no distinctions between the CK, 10 μmol·L^−1^, and 30 μmol·L^−1^ treatments. Interestingly, there was a remarkable increase in the number of root forks following melatonin treatment, particularly in the 10 μmol·L^−1^ and 30 μmol·L^−1^ treatments. Conversely, the number of root tips decreased, with no significant variation observed in the number of crossings.

### 2.3. Effects of Exogenous Melatonin Treatment on Root Hair Length and Density of Tomato Seedlings

Following melatonin treatment of tomato seedling roots, root hairs exhibited consistent alterations. As shown in Figure 3, both primary root hairs (Figure 3A) and lateral root hairs (Figure 3B) exhibited an initial increase in length and density with rising melatonin concentration, followed by a decrease. The most pronounced effect was observed in the 10 μmol·L^−1^ treatment. Notably, in the 30 μmol·L^−1^ and 50 μmol·L^−1^ treatments, melatonin exhibited a pronounced inhibitory effect on root hairs, leading to significantly reduced length and density compared with the CK and 10 μmol·L^−1^ treatments.

### 2.4. Effects of Exogenous Melatonin Treatment on Root Meristem Cells of Tomato Seedlings 

Alterations were observed in both the cell size and cell number within the meristem of tomato seedling roots following treatment with varying melatonin concentrations. When compared with the CK treatment (Figure 4A), the 10 μmol·L^−1^ treatment (Figure 4B) exhibited an increase in the size of meristem cells, accompanied by an increase in the number of meristem cells. Notably, following the elevation in melatonin concentration, the number of meristem cells in both the 30 μmol·L^−1^ (Figure 4C) and 50 μmol·L^−1^ treatments (Figure 4D) remained higher than in the CK treatment, although the size of meristem cells was smaller.

### 2.5. Effects of Exogenous Melatonin Treatment on Root Hormone Content of Tomato Seedlings

Figure 5 illustrates significant alterations in the root hormone content of tomato seedlings following treatment with varying melatonin concentrations. Zeatin (ZT) content initially increased and then decreased with rising melatonin concentrations, peaking in the 30 μmol·L^−1^ treatment. In comparison with the CK treatment, the 10 μmol·L^−1^, 30 μmol·L^−1^, and 50 μmol·L^−1^ treatments exhibited increases of 225.7%, 1031%, and 189.6%, respectively (Figure 5A). However, as another cytokinin, there were no significant differences in the content of trans-zeatin (TZR) nucleoside between the CK, 10 μmol·L^−1^, and 30 μmol·L^−1^ treatments. In contrast, the content of trans-zeatin nucleoside in the 50 μmol·L^−1^ treatment showed a sudden decrease that was significantly lower than in the other treatments (Figure 5B). The change in gibberellin 3 (GA3) content followed a similar pattern to zeatin, displaying an initial increase followed by a decrease. However, the notable difference was that the content of GA3 in tomato roots was substantially higher than that of zeatin. The 10 μmol·L^−1^, 30 μmol·L^−1^, and 50 μmol·L^−1^ treatments exhibited increases of 68.3%, 216.8%, and 134.8%, respectively, compared with the CK treatment (Figure 5C). As the melatonin concentration increased, the auxin content likewise increased. No significant difference was observed between the CK and 10 μmol·L^−1^ treatments. However, the 30 μmol·L^−1^ and 50 μmol·L^−1^ treatments exhibited significantly higher levels than the former two. Notably, the 50 μmol·L^−1^ treatment displayed the most substantial increase, being 77.4%, 70.8%, and 21.3% higher than the CK, 10 μmol·L^−1^, and 30 μmol·L^−1^ treatments, respectively (Figure 5D). Intriguingly, the alteration in abscisic acid (ABA) content exhibited a parallel pattern to that of auxin, showing an increase with rising melatonin concentration (Figure 5E). The variation in brassinolide (BR) displayed an initial increase followed by a decrease, with its peak occurring in the 10 μmol·L^−1^ treatment. Notably, as the melatonin concentration increased, the change in brassinolide was more pronounced, and the 30 μmol·L^−1^ and 50 μmol·L^−1^ treatments exhibited lower levels than the CK treatment (Figure 5F).

### 2.6. Effects of Exogenous Melatonin Treatment on the Expression of Root-Related Genes in Tomato Seedlings

As shown in Figure 6, following treatment of tomato seedling roots with varying melatonin concentrations, firstly, tomato lateral root-related genes including *SlCDKA1* (Figure 6A), *SlCYCA3;1* (Figure 6C), *SlARF2* (Figure 6E), and *SlF3H* (Figure 6F) exhibited a consistent trend. They increased with the rise in melatonin concentration, with the 50 μmol·L^−1^ treatment showing significantly higher levels than the other treatments. However, it is noteworthy that there were no significant differences between the CK treatment and the 10 μmol·L^−1^ treatment in the case of *SlCDKA1* and *SlCYCA3;1*. Additionally, there were no significant differences between the 10 μmol·L^−1^ treatment and 30 μmol·L^−1^ treatment for *SlARF2*, but significant differences were observed among the four treatments in the case of *SlF3H*. Conversely, following treatment with varying melatonin concentrations, *SlCYCA2;1* (Figure 6B) exhibited significantly lower levels in the 50 μmol·L^−1^ treatment compared with the other treatments, whereas *SlKRP2* (Figure 6D) in the 30 μmol·L^−1^ treatment displayed significantly higher levels than the other treatments. Conversely, the relative expression levels of tomato root hair-related genes, including *SlExt1* (Figure 6G) and *SlKT1* (Figure 6H), exhibited an initial increase followed by a decrease, reaching their peak during the 10 μmol·L^−1^ treatment.

## 3. Discussion

In this experiment, hydroponically grown tomato plants were exposed to various concentrations of exogenous melatonin. Simultaneously, we demonstrated certain mechanisms through which melatonin regulates the growth and development of tomato roots, focusing on tomato root development-related genes and endogenous hormones. The results indicated that exogenous melatonin had the capability to alter root structure, control root hair growth, and influence meristem cell development. These effects were mediated by melatonin-related genes associated with tomato root growth and development, and they worked in conjunction with other endogenous hormones. Therefore, the authors initially examined the impact of exogenous melatonin on tomato root development and subsequently delved into the underlying mechanisms by which exogenous melatonin affects tomato root development.

The accumulation of plant biomass is used as an indicator to evaluate the degree of healthy growth and development in plants [53]. ‘TTC’-measured root activity reflects the dehydrogenase activity within the root system and indirectly signifies the extent of root development [54]. The study found that the application of exogenous melatonin at low concentrations (less than 30 μmol·L^−1^) led to the increased accumulation of tomato root biomass and enhanced root activity. Notably, the most pronounced effect was observed within the range of 10–30 μmol·L^−1^. This conclusion aligns with earlier research by Zhang et al. [55] and Liu et al. [56]. Conversely, exposure to high melatonin concentrations (melatonin concentrations exceeding 50 μmol·L^−1^) resulted in significant damage to root development. This harm can be attributed to the toxic effects of high melatonin concentrations on tomato roots. In order to gain a more profound insight into the influence of exogenous melatonin on tomato roots, we analyzed the cell morphology of the tomato root meristem. The results showed that when exposed to low concentrations of exogenous melatonin (0–10 μmol·L^−1^), tomato meristem cells notably enlarged in size, and the quantity of meristem cells also increased. In contrast, at high concentrations of exogenous melatonin (30–50 μmol·L^−1^), the size of tomato meristem cells decreased, while their numbers increased. Therefore, the changes in root biomass caused by exogenous melatonin could be ascribed to alterations in cell morphology.

The root structure of plants primarily consists of the primary root, lateral roots, and root hairs. Lateral roots assist in nutrient absorption from the surrounding environment, and root hairs are vital tissues for absorbing nutrients and water [57]. Wang et al. [58] discovered that concentrations of exogenous melatonin of less than 0.1 μmol·L^−1^ promoted the growth of primary roots in Arabidopsis thaliana, whereas concentrations exceeding 1 μmol·L^−1^ hindered primary root elongation. It is evident that melatonin affects the development of primary roots in both tomato and Arabidopsis thaliana, but further verification is needed to confirm the phenotype at melatonin concentrations of less than 0.1 μmol·L^−1^. In this experiment, it was observed that the number of lateral roots in tomatoes increased as the exogenous melatonin concentration increased. This phenomenon was pronounced at melatonin concentrations of 10–30 μmol·L^−1^ but was not evident at concentrations exceeding 50 μmol·L^−1^. The lack of an observable effect at higher concentrations may be attributed to potential root damage caused by high melatonin levels. The effect of melatonin on the development of lateral roots and adventitious roots has been a widely studied topic in recent decades. Previous studies [31,34] have shown that an appropriate melatonin concentration will increase the number of lateral roots and adventitious roots. The findings of this study align with those of previous research studies. Nevertheless, the mechanism behind melatonin-induced lateral root formation remains unclear. Consequently, the authors identified genes associated with the formation and development of tomato lateral roots for further investigation. *CDKA1*, *CYCA2;1*, *CYCA3;1*, and *KRP2* genes are regulators of the cell cycle. Substantial evidence indicates that cell cycle activation is a crucial mechanism in lateral root formation [59,60,61]. Previous genetic and molecular evidence also suggests that *CDKA1*, *CYCA3;1*, and *KRP2* are significant molecular markers associated with lateral root initiation [62,63,64]. *ARF2* is a gene associated with auxin signal transduction. Previous research has indicated a connection between *ARF2* and the formation of lateral roots [65], while *F3H* represents the initial step in flavonol synthesis. Previous studies have demonstrated that overexpressing this gene results in an increased number of lateral roots, indicating a positive role of *F3H* in lateral root development [66]. This study discovered that exogenous melatonin markedly influenced the expression of *SlCDKA1*, *SlCYCA3;1*, *SlARF2*, and *SlF3H* genes in roots, with their expression levels rising alongside melatonin concentration. This suggests that exogenous melatonin promotes the growth of tomato lateral roots, possibly through the mediation of *SlCDKA1*, *SlCYCA3;1*, *SlARF2*, and *SlF3H*. However, the precise mediation mechanism requires further verification using modern molecular techniques. Root hairs are a crucial component of the root structure. This study demonstrated that at melatonin concentrations ranging from 0 to 10 μmol·L^−1^, both the number and length of tomato root hairs increased with the rising melatonin concentration. Intriguingly, at melatonin concentrations exceeding 30 μmol·L^−1^, there was a sharp reduction in both the number and length of tomato root hairs. This indicates that melatonin adheres to the ‘low concentration promotes, high concentration inhibits’ principle in root hair growth. To explore the potential mechanism behind changes in root hair development, the author examined genes related to tomato root hairs. The *SlExt1* gene encodes a cell wall elongation protein in tomato roots and plays a role in the growth of root hair cell tips during root hair development [59]. The *SlKT1* gene is a crucial regulator of K^+^ channels in root hairs [60], and its expression level serves as an indicator of root hairs’ ability to absorb K^+^. The results indicated that the expression of *SlExt1* significantly increased at 10 μmol·L^−1^ melatonin but decreased as melatonin concentrations exceeded 30 μmol·L^−1^. This suggests that the melatonin-mediated gene *SlExt1* may be responsible for the increase in tomato root hairs induced by low concentrations of exogenous melatonin (0–10 μmol·L^−1^). Simultaneously, the expression pattern of the *SlKT1* gene paralleled that of *SlExt1*, suggesting that low concentrations of exogenous melatonin (0–10 μmol·L^−1^) could augment the absorption capacity of root hairs for beneficial ions.

In this experiment, it was observed that exogenous melatonin treatment led to alterations in several hormones within tomato roots. Initially, the auxin content increased in response to rising melatonin concentrations. It has been reported that melatonin treatment resulted in elevated endogenous IAA content compared with untreated plants, as observed in mustard [38] and tomatoes [37]. This suggests that melatonin and auxin collectively play a role in regulating the alterations in tomato roots. Subsequently, the zeatin and gibberellin 3 content initially increased and then decreased with rising melatonin concentrations in this experiment. The authors hypothesize that the alteration in tomato root morphology may be linked to the fluctuation in zeatin and gibberellin 3 content. Nevertheless, prior studies have demonstrated that exogenous melatonin can collaborate with cytokinin to regulate plant photosynthesis [61] and provide mechanistic defense against heat stress [67]. Moreover, melatonin-induced elevation of gibberellin 3 content has been shown to enhance seed germination under salt stress [68] and promote the formation of lateral roots [69]. Consequently, there has been limited research on the regulation of tomato root morphology by zeatin and gibberellin 3, which could be a focal point for future investigations. Interestingly, prior studies have revealed that various hormones can influence the morphology of root hairs. Auxin [70] and cytokinin [71] have a positive influence on root hair growth, while abscisic acid [72] and brassinolide [73] have a negative impact on root hair growth. In summary, biological processes are intricate, and the alterations in tomato root morphology induced by exogenous melatonin are no exception. Consequently, the modification of tomato root architecture and root hair growth is governed by melatonin in conjunction with auxin, cytokinin, gibberellin, and abscisic acid.

## 4. Materials and Methods

### 4.1. Plant Material and Growth Conditions

The experimental material used in this study was the ‘184’ tomato variety. Tomato seeds of uniform size and full integrity were subjected to a 5 min disinfection in a 2% sodium hypochlorite solution, followed by thorough rinsing with copious amounts of water. The seeds were agitated on a shaker at 28 °C until germination occurred, after which they were transferred to vermiculite for one week of growth. Subsequently, tomato seedlings with robust root systems were carefully chosen and transplanted into hydroponic containers. Every hydroponic container was filled with 10 L of reverse osmosis (RO) water and placed in a growth chamber with controlled temperature (28 ± 1 °C). The light intensity was maintained at 250 μmol·m^−2^·s^−1^ with a photoperiod of 14 h of light and 10 h of darkness.

### 4.2. Experimental Design

Melatonin treatment was administered at the stage of cotyledon flattening and the emergence of the first true leaf. The experimental treatments employed in this study comprised the following: a CK (control) treatment, where seedlings were grown in distilled water for a duration of seven days; a 10 μmol·L^−1^ treatment, where seedlings were grown in a 10 μmol·L^−1^ melatonin solution for seven days; a 30 μmol·L^−1^ treatment, where seedlings were grown in a 30 μmol·L^−1^ melatonin solution for seven days; and a 50 μmol·L^−1^ treatment, where seedlings were cultivated in a 50 μmol·L^−1^ melatonin solution for seven days. On the eighth day, root samples were collected from the junction between the root and shoot.

### 4.3. Fresh Weight and Root Activity

For each treatment, three tomato seedling roots with uniform growth were chosen, and any adhering water on the roots was removed before measuring their fresh weight using an analytical balance.

Root activity was determined by measuring respiratory activity using 2, 3, 5-triphenyl tetrazolium chloride (TTC) as per the procedure described by Yingdui He et al. [54]. Approximately 0.5 g of root tissue was introduced into a 0.4% TTC solution along with an equal volume of phosphate buffer (10 mL), and this mixture was incubated in the dark at 37 °C for 1 h. Subsequently, 2 mL of 1 mol·L^−1^ H2SO4 was added to initiate the reaction. The roots were ground in ethyl acetate to extract the reduced red tetrazolium, and the absorbance at 485 nm was determined using a spectrophotometer. Based on the A485 value of the sample, X (mg) was calculated using a standard curve or regression equation, and the root activity, indicating the strength of TTC reduction by the roots, was computed using the following formula: TTC reduction intensity [mg·(g·h)^−1^] =“X”/”W·t”
[W: fresh weight of the sample (g); T: reaction time (h)]

### 4.4. Root Scanning

The determination of root architecture parameters followed the method by Tripathi et al. [74]. Three tomato seedlings with uniform growth were chosen for each treatment, and their roots were gently spread out in a transparent tray using tweezers. The roots of each treated tomato were subjected to scanning using a root scanner from Regent Instruments, Inc. (Quebec City, QC, Canada), and the resulting scanned images were preserved. Subsequently, the root images were subjected to analysis using specialized root analysis software (Win RHIZO Pro LA2400, Regent Instruments Inc., Quebec City, QC, Canada) to extract data on the root length, total root surface area, total root volume, number of lateral roots, number of root tips, and number of crossings.

### 4.5. Tomato Seedling Root Hair Length and Density

The determination of root hair length and density in tomato seedlings followed the method by Cheng Fang et al. [75] with slight modifications. Following melatonin treatment, 1 cm segments were excised from the root tips of the primary and first lateral roots of each seedling, and these root segments were subsequently fixed in 70% ethanol. The fixed root segments were positioned on a glass slide, with each segment being immersed in a small droplet of water for microscopic examination. Root hairs were examined under a digital stereo microscope (OLYMPUS SZX12 (Olympus, Tokyo, Japan)) with the assistance of the ‘Mshot Image Analysis System 1.1.4’ software.

### 4.6. The Size and Number of Meristematic Cells

The size of the root meristem and meristem cells was determined by the conventional paraffin section method. (1) The root samples were fixed with 40% paraformaldehyde in a 2 mL sampling tube and placed overnight under vacuum pressure. (2) The fixed sample was dehydrated in graded ethanol (30%, 50%, 70%, 80%, 90%, 95%, 100%) for 15 min. (3) Dehydrated samples were made transparent using a xylene/ethanol series (xylene:ethanol = 1:3, xylene:ethanol = 1:1, xylene:ethanol = 3:1, 100% xylene) for 2 h each. In particular, 100% xylene was made transparent twice, the last time being overnight. (4) The samples were waxed with a paraffin/xylene series (paraffin:xylene = 1:3, paraffin:xylene = 1:1, paraffin:xylene = 3:1, 100% paraffin) for 2 h each. Similarly, they were waxed with 100% paraffin twice, the last time being overnight. (5) The samples were embedded in paraffin by a paraffin embedding machine (HistoCore H-C). After freezing at −20 °C for 15 min, the samples were taken out, and the mold was removed. (6) The sample was longitudinally sliced using a fully automatic paraffin slicing machine (HistoCore AUTOCUT), and a paraffin strip with a thickness of 10 μm was stretched in a water bath at 37 °C. The slices were mounted on slides and baked in an oven at 37 °C for three days. (7) Xylene dewaxing was performed twice every 5 min and alcohol dewaxing twice every 2 min after drying in the fume hood. (8) Finally, it was dyed with a 0.5% toluidine blue solution for 5 min and was washed and dried after dyeing to be tested on the machine.

The paraffin sections of roots were observed and photographed using a positive and negative integrated fluorescence microscope (Revolve RVL-100-G) and using Revolvepro v2023 image acquisition software.

### 4.7. Hormone Content

The determination of hormone content was determined by HPLC with reference to Dobrev and Vankova’s [76] method, and some modifications were performed.

The 0.5 g of root tissue was quickly ground into powder by a freeze grinder, transferred to a 10 mL centrifuge tube, and added to 5 mL of the extract (n-propanol: distilled water:hydrochloric acid = 2:1:0.002, volume ratio). Then, we shook the shaker at 25 °C at 200 g for 30 min, removed the centrifuge tube, added 2 mL dichloromethane, shook again for 30 min, and centrifuged at 18 °C at 13,000× g for 10 min. Then, 2 mL of the supernatant was taken and placed in a 5 mL centrifuge tube, placed in a vacuum centrifuge concentrator, concentrated to full dryness at 30 °C, and then taken out and had 1 mL of 80% methanol solution added to it to dissolve. Finally, the extract was passed through a 0.22 μm organic filter membrane in a brown chromatographic bottle for detection (chromatographic conditions: Agilent 1100 high-performance liquid chromatograph; Agilent, USA). Chromatographic column: ZORBAX SB-C18 (4.6 × 250 mm, 5 μm); column temperature: 30 °C; and mobile phase: methanol and 0.1% phosphoric acid (*v*:*v* = 1:9). The flow rate was 1.0 mL·min^−1^. The detection wavelength was 254 nm. The injection volume was 10 μL.

The standard curve was calculated according to the concentration and peak area of each hormone standard. Then, according to the peak area of the sample, the X (μg·mL^−1^) was calculated according to the standard curve, and then, the content of each hormone in the root system was calculated according to the following formula:Hormone content (μg·g^−1^) = “X·1”/”0.5”

### 4.8. Quantitative Real-Time PCR

Quantitative real-time PCR was based on the method of Muhammad Ahsan Altaf [77], and some modifications were performed. According to the manufacturer’s instructions, total RNA was extracted from different treated tomato roots using an RNA simple total RNA extraction kit (TIANGEN, Beijing, China). With the help of agarose gel electrophoresis and a K5800 micro spectrophotometer (KAIAO, Beijing, China), the purity of the extracted RNA was detected. The RNA was then reverse transcribed using the HiScript II Q RT SuperMix for qPCR (+gDNA wiper) reverse transcription kit (Vazyme, Nanjing, China) for complementary DNA (cDNA) synthesis. For the qRT-PCR analysis, cDNA was used as a template, and the SYBR^®^GREEN Premix Pro Taq HS qPCR Kit (ROX Plus) (Accurate Biotechnology, Changsha, China) was used in the QuantStudio5 P-qPCR system (Applied Biosystems, Waltham, MA, USA), using 96-well plates for the qRT-PCR.

The detailed information on the primers used in this study is provided in Table 2, and actin is used as the reference gene. The relative expression level of the gene was calculated using the 2^−ΔΔCt^ method by referring to the Livak [78] and Schmittgen [79] method.

### 4.9. Statistical Analysis

One-way analysis of variance and Duncan’s multiple range tests (*p* < 0.05) were used for data analysis in SPSS (version 22.0; SPSS Institute Inc., Chicago, IL, USA) software. We produced tables and processed the data using Microsoft Excel 2016 (Microsoft Inc., Redmond, WA, USA) and drew diagrams using Origin 2021 (Origin, Inc., San Francisco, CA, USA).

## 5. Conclusions

This study found that exogenous melatonin at a concentration of 10–30 μmol·L^−1^ can significantly promote tomato lateral root development and root hair growth, thereby increasing tomato root biomass accumulation and root activity. Further research found that exogenous melatonin changes tomato root morphology by affecting ZT, GA3, IAA, ABA, and BR endogenous hormones and *SlCDKA1*, *SlCYCA3;1*, *SlARF2*, *SlF3H*, and *SlKT1* genes. Consequently, this study offers additional evidence that supports melatonin’s role in regulating plant growth and development, and it establishes the groundwork for investigating the mechanisms through which melatonin influences root morphological changes.

## Figures and Tables

**Figure 1 plants-13-00383-f001:**
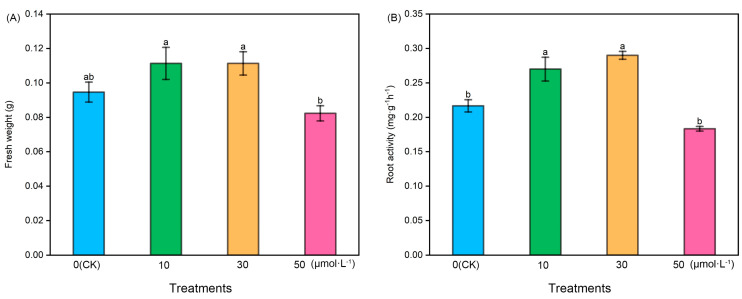
The fresh weight and root activity index of tomato seedlings under the treatment of melatonin. (**A**) Fresh weight and (**B**) root activity. Different letters indicate significant differences between treatments (*p* < 0.05).

**Figure 2 plants-13-00383-f002:**
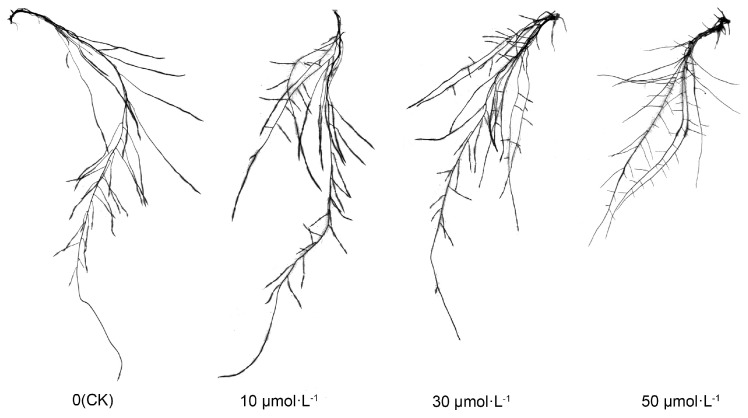
Visual display of tomato seedling roots under melatonin treatment.

**Figure 3 plants-13-00383-f003:**
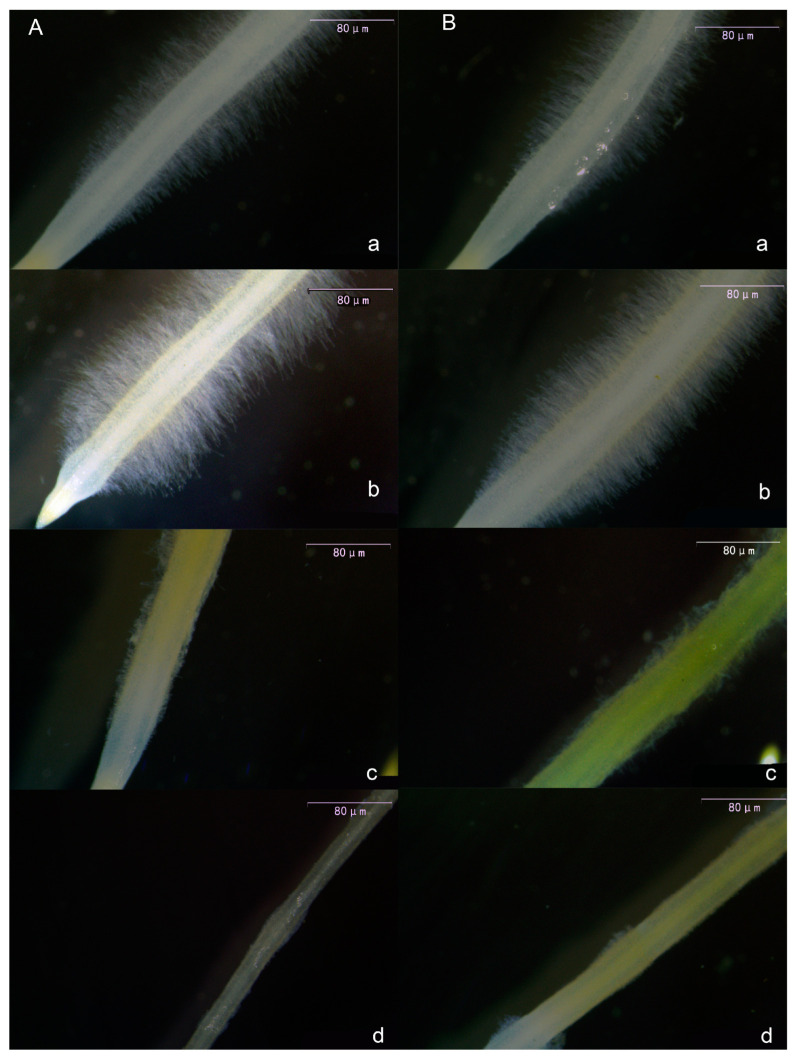
Visual display of the effect of melatonin on the main root hair (**A**) and lateral root hair (**B**) of tomato seedlings. (a) CK treatment, (b) 10 μmol·L^−1^ treatment, (c) 30 μmol·L^−1^ treatment, and (d) 50 μmol·L^−1^ treatment.

**Figure 4 plants-13-00383-f004:**
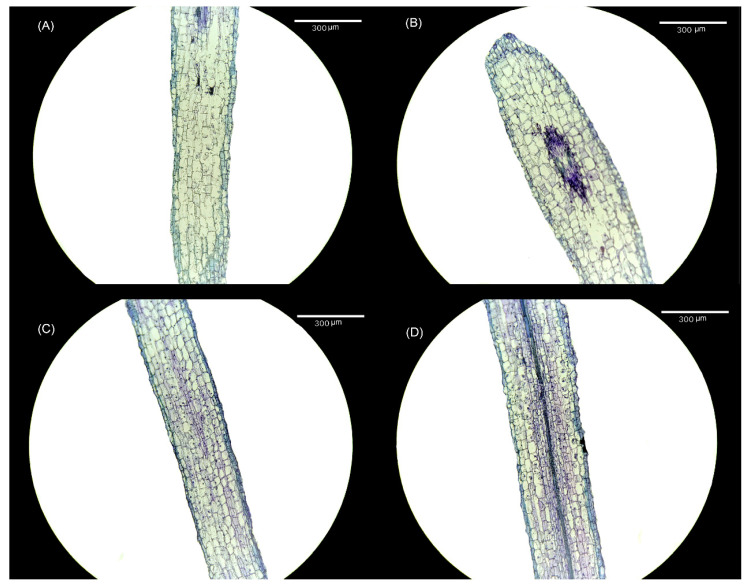
Visual display of the effect of melatonin treatment on root meristem cells of tomato seedlings. (**A**) CK treatment, (**B**) 10 μmol·L^−1^ treatment, (**C**) 30 μmol·L^−1^ treatment, and (**D**) 50 μmol·L^−1^ treatment.

**Figure 5 plants-13-00383-f005:**
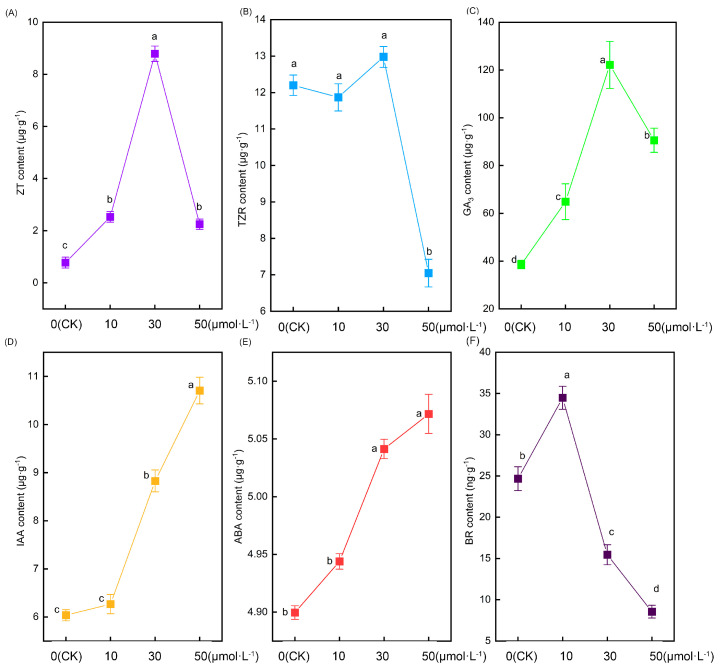
The root hormone content indexes of tomato seedlings under the treatment of melatonin. (**A**) Xeatin, (**B**) trans-zeatin nucleoside, (**C**) gibberellin 3, (**D**) auxin, (**E**) abscisic acid, and (**F**) brassinosteroids. Different letters indicate significant differences between treatments (*p* < 0.05).

**Figure 6 plants-13-00383-f006:**
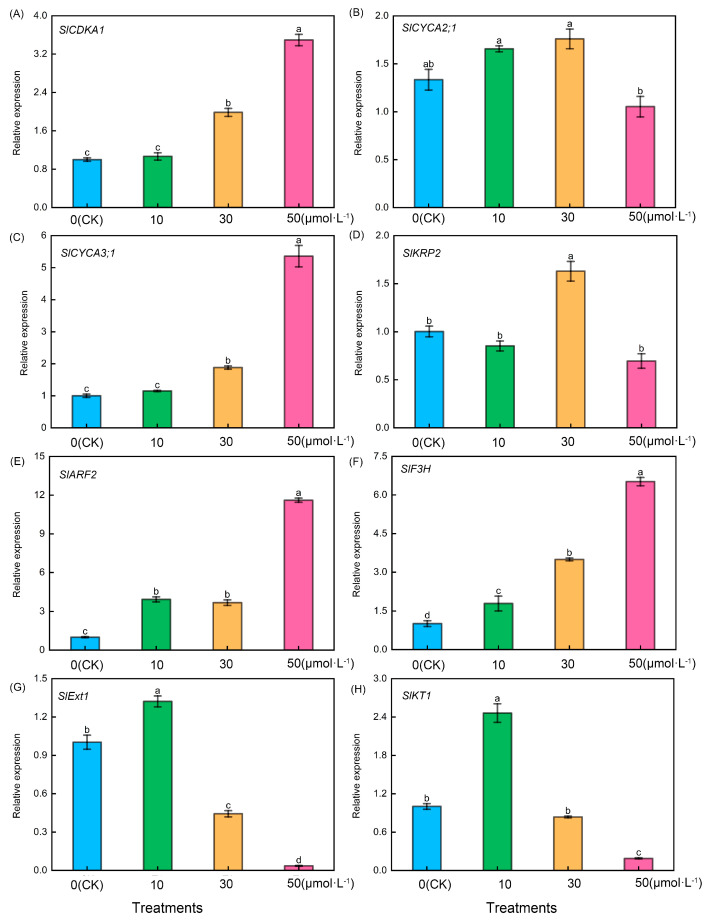
Expression levels of root-related genes in tomato seedlings under melatonin treatment. Lateral root development-related genes: (**A**) *SlCDKA1*, (**B**) *SlCYCA2;1*, (**C**) *SlCYCA3;1*, (**D**) *SlKRP2*, (**E**) *SlARF2*, and (**F**) *SlF3H*. Root hair-related genes: (**G**) *SlExt1* and (**H**) *SlKT1*. Different letters indicate significant differences between treatments (*p* < 0.05).

**Table 1 plants-13-00383-t001:** Root architecture parameters of tomato seedlings under melatonin treatment.

Treatments	Total Root Length·cm^−1^	Root Volume·cm^−3^	Root Surarea cm^−2^	Root Forks	Root Tips	Root Crossings
0 (CK)	102.77 ± 2.27 ^a^	0.11 ± 0.00 ^a^	11.71 ± 0.06 ^a^	220 ± 17.9 ^b^	412 ± 9.24 ^a^	41.67 ± 3.76 ^a^
10 μmol·L^−1^	97.19 ± 1.40 ^a^	0.14 ± 0.02 ^a^	12.49 ± 1.21 ^a^	367 ± 13.57 ^a^	352 ± 16.46 ^b^	45.33 ± 3.76 ^a^
30 μmol·L^−1^	73.88 ± 2.09 ^c^	0.11 ± 0.01 ^a^	11.84 ± 0.56 ^a^	349 ± 28.00 ^a^	283 ± 16.17 ^c^	48.00 ± 5.20 ^a^
50 μmol·L^−1^	81.87 ± 0.64 ^b^	0.10 ± 0.02 ^a^	9.26 ± 0.18 ^b^	269 ± 13.86 ^b^	249 ± 2.91 ^c^	52.00 ± 6.93 ^a^

The mean and SE values were calculated from at least three independent experiments. According to Duncan’s multi-range test, lowercase letters in the table represent different levels of difference(*p* < 0.05).

**Table 2 plants-13-00383-t002:** The accession numbers and primer sequences for qPCR.

Primer Names	Accession Number	Sequences (5′→3′)
*SlActin*	NM_001330119.1	F: CCACGAGACTACATACAA
		R: TACCACCACTGAGCACAA
*SlCYCA3;1*	NM_001247858.2	F: TGCGGTTCTTGCCATCA
		R: CGCCCAGTTGCTTCCA
*SlCYCA2;1*	NM_001246839.2	F: CATTAACAAGGGTATGCGAA
		R: GTCAGGTAAAGAGTGTCCGG
*SlCDKA1*	NM_001247447.2	F: CACTTGCCTGTCGCCTCCTC
		R: ACCCCCTCGTCTTCCTGCTC
*SlKRP2*	NM_001247055.2	F: CTTCACAAACCACCCACCCC
		R: TTTCGTCCACCTCCCTCACC
*SlARF2*	XM_010320115.3	F: CTATGCCGTGTTGTGAATGTCCTG
		R: ACCGTGAGTGCTTGTATCAGAGG
*SlF3H*	NM_001374424	F: TGAAAAGACCCTTGAAACAA
		R: CGATTCTCTCACATATTTCA
*SlExt1*	NM_001247899.3	F: AAGAGCTATGAGC TCCCAGATGG
		R: TTAATCTTCATG CTGCTAGGAGC
*SlKT1*	NM_001247329.3	F: GAGGTCAGGGCTGGTGATCTTTG
		R: TGGCACAGTCTCTTCGTTCGTAC

## Data Availability

Data are available from the corresponding author.

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
