# Peer review of "Melatonin Modulates Tomato Root Morphology by Regulating Key Genes and Endogenous Hormones"

_plants, 2024, doi:10.3390/plants13030383_

Round 1
Reviewer 1 Report
Comments and Suggestions for Authors
the author address following suggession
start one more line, start of the abstract
Revise conclusion lines at the end of abstract
add more specific keywords
Line 66-67: botanical name should be italic
Line 71: Botanical name should be italic
Line 73: Arabdopsis should be italce
Line 103-104: objective not well written
in the figure unit not suitable
Line 341: remove typo, superscript m-2·s-1"
overall manuscript written well,, but i have some question,
How melatonin solution was prepared for this study?
How author choose melatonin level? usually for tomato researcher used 100 micromole for this study,
Line 368-374: provide suitable reference for this study
revise manuscript conclusion
Reviewer 2 Report
Comments and Suggestions for Authors
The reviewed work concerns the very interesting problem of the role of melatonin in plant growth and development. Exogenous melatonin was added to hydroponic tomato cultivation. Root growth and morphology as well as endogenous hormones and relevant genes were taken into account. A clear effect of melatonin on the growth, morphology, hormone levels and gene expression in tomato was found. In the introduction, the role of roots and morphological diversity in the life of the plant was described. Then, the role of melatonin in animals and the discovery and role of melatonin in plants are presented. Tomato has also been described as an important plant in agricultural crops.
The results show the effect of melatonin on the fresh weight and root activity, root architecture parameters, root hair length and density, root meristem cells, root hormone content, root-related genes. The research results were clearly presented and analysed statistically. The discussion is very extensive, takes into account all factors presented in the work and refers properly to the existing literature. The conclusions of the work are appropriate for the research presented. Cited literature is properly selected.
line 163, page 5:
“2.4. Effects of exogenous melatonin treatment on root meristem cells of tomato seedlings”
Words should start with a capital letter, as in the other subheadings.
In the literature list, some journal names are written in lower case but should be capitalized, e.g. in position: 1, 3, 6, .....24, 25, 29, 30, 36, 37, 38, ...... .
Round 2
Reviewer 1 Report
Comments and Suggestions for Authors
accepted